# Ubiquitous Expression of *mPolg^mut^* Leads to the Accumulation of Cytotoxic CD8+ T Lymphocytes in Young Mice

**DOI:** 10.3390/life15121863

**Published:** 2025-12-05

**Authors:** Nadezhda A. Kuznetsova, Ksenia K. Kochegarova, Iuliia P. Baikova, Eugenii N. Korshunov, Leonid A. Ilchuk, Marina V. Kubekina, Alexandra V. Bruter, Yulia Yu. Silaeva, Ekaterina A. Varlamova

**Affiliations:** Institute of Gene Biology, Russian Academy of Sciences, Vavilov St., 34/5, 119334 Moscow, Russia

**Keywords:** DNA polymerase γ, aging, mutator mice, immunity, T lymphocytes

## Abstract

Age-related changes are associated with mitochondrial dysfunction, which is often caused by the accumulation of mutations in mitochondrial DNA (mtDNA). One common model of aging and age-related diseases involves mice with a mutant DNA polymerase γ (PolG^mut^) whose proofreading function is impaired, which leads to the accumulation of mutations in mtDNA. The main limitation of such a model is that introducing a mutation into the mouse’s own gene leads to the accumulation of mutations in mtDNA over several generations, making it impossible to rule out whether mtDNA mutations or compensatory effects are the cause of functional impairments such as accelerated aging. This paper describes two lines of transgenic animals with inducible expression of PolG^mut^. This inducible system prevents mutation accumulation in the germline, promoting stable reproduction and reproducibility of mice, increasing experimental flexibility for various studies of mitochondrial diseases. PolG^mut^ activation at different stages of life and different tissues allows us to study the progression of pathological changes during mitochondrial aging over time and detect the onset of mutation accumulation. The simplicity, reproducibility, and temporal control of this system represent a significant methodological improvement for studying mitochondrial mutagenesis and the pathophysiology of aging. Using this model, we demonstrated that the most pronounced pathology in these animals is accelerated thymus involution and the accumulation of cytotoxic effector CD8+ T cells in the peripheral immune organs, while no significant abnormalities were observed in other organs and systems. These data probably indicate that mtDNA mutations primarily impair T-cell immune function.

## 1. Introduction

Mitochondrial diseases are a heterogeneous group of pathological conditions associated with mitochondrial dysfunction. The degree of mitochondrial dysfunction is determined by the proportion of defective mitochondrial DNA (mtDNA) in the cell (the level of heteroplasmy), as well as the localization of dysfunctional mitochondria. The wide range of clinical manifestations of mitochondrial dysfunction complicates the study of pathogenesis processes and necessitates the creation of adequate model systems.

Numerous mitochondrial disorders can be caused by mutations in the nuclear gene *POLG* [1], that encodes DNA polymerase γ (PolG). This enzyme has DNA polymerase and 3′-5′ exonuclease activity and, thus, plays a major role in mitochondrial DNA replication and repair. Mutations in the *POLG* gene are associated with diseases such as progressive external ophthalmoplegia, Alpers syndrome, childhood myocerebrohepatopathy, Parkinson’s disease, MELAS syndrome, etc. (see [2] for review).

Mice with mutations in the DNA polymerase γ gene (*Polg*, here *mPolg* for clarity) are a well-studied model of the progeroid phenotype [3,4,5]. The main tool for studying the role of *mPolg* mutations in aging and mitochondrial diseases are genetically modified mice with a substitution of aspartic acid at position 257 to alanine (D257A), which disables the proofreading activity of PolG—PolG^D257A^ mice [3,4]. These model mice have become a reference for studying the mechanisms of mitochondrial dysfunction and premature aging [6,7], which is expressed in a decrease in endurance, muscle mass [8,9,10], impaired reproductive function [11], immune dysfunction [12,13] and a reduction in lifespan [3,12,14]. The main limitation of the classical model is the accumulation of mtDNA mutations over a number of generations. As a result, investigational data may be confounded by secondary compensatory effects, complicating the interpretation of the impact of primary mtDNA mutations on aging phenotypes. Furthermore, mutation accumulation occurs independently from generation to generation, potentially leading to wide data scatter and biased interpretation. In the present study, *transgenic* mice with inducible expression of a mutant variant of the *mPolg* gene were used. The advantage of such models is that expression of the mutant gene is only induced in the offspring upon crossing transgenic mice with an activator line promptly or upon a later Cre-mediated excision of the STOP cassette in adult animals, thus providing high flexibility in temporal and system-specific manner of activation, should they be required. We demonstrated that mitochondrial dysfunction primarily affects the immune system in mice, with the most significant impact on the T-cell component of immunity. In this study, we showed that expression of the mutant form of *mPolg* de novo leads to accelerated thymus involution in young mice and a change in the ratio of CD4+/CD8+ T-cell populations in the peripheral lymphoid organs of adult animals, but does not lead to pronounced pathological changes in the muscular and reproductive systems. The phenotype of the PolG^mut^ mice differs from the more aggressive pathologies observed in existing PolG^D257A^ “mutator” mice [3,4], but we believe our model is better suited for studying immunological disorders and immune senescence caused by mitochondrial diseases.

## 2. Materials and Methods

### 2.1. Animals

The study protocol was approved by the Ethics Committee for Animal Research of the Institute of Gene Biology of the Russian Academy of Sciences (protocol code 26, from 15 December 2023). The studies were conducted on mice with the C57BL/6 genetic background (haplotype H-2K^b^). The activator lines CMV-Cre and ROSA-Cre/ER^T2^ [15] were used in the work, as well as transgenic mice of the PolG^mut^-STOP-GFP [16] and PolG^mut^-STOP-pKB2 lines, which have not been described previously (Appendix A, Table A1). To induce transgene expression, PolG^mut^-STOP-GFP and PolG^mut^-STOP-pKB2 mice were crossed with activator strain mice; the presence of both transgenes was determined in the offspring using PCR, as described in [16] (Appendix A). In transgenic animals obtained by crossing PolG^mut^-STOP-GFP or PolG^mut^-STOP-pKB2 with the CMV-Cre activator line, expression of the PolG^mut^ begins upon activation of the zygotic genome. Mice obtained by crossing PolG^mut^-STOP-GFP or PolG^mut^-STOP-pKB2 lines to the ROSA-Cre/ER^T2^ activator line require tamoxifen induction to activate the transgene. For tamoxifen induction, a solution of tamoxifen (Sigma-Aldrich, Burlington, MA, USA) in corn oil (Sigma-Aldrich, Burlington, MA, USA) was prepared and injected at a dose of 10 mg per mouse as described in [15]. During the experimental work, the animals were kept in the vivarium of the Core Facility of the Institute of Gene Biology of the Russian Academy of Sciences with constant access to water and food. The light cycle was 13/11, air temperature 23 ± 1 °C, humidity 42 ± 5%. CMV-Cre or ROSA-Cre/ER^T2^ siblings of the same age and sex as transgenic mice, respectively, were used as control animals. Male mice were used for all experiments except histological analysis.

### 2.2. Physiological Tests

Physiological tests of strength and endurance were conducted on males of 4 to 10 months of age, as described in [16]. Each series of physiological tests was carried in three replicates, separated by a one-week interval. The break between series was two weeks. The time interval between endurance and grip strength tests was at least 24 h.

### 2.3. Study of Mouse Thymus Morphology

Young 4-month-old male mice were sacrificed, and their thymuses were removed for morphological assessment. Images were acquired under an SMZ800N binocular (Nikon, Tokyo, Japan) using a Basler PowerPack microscopy camera (“Basler AG”, Ahrensburg, Germany).

### 2.4. Isolation of Mouse Spleen or Thymus Cells

Male mice spleens or thymuses were placed in cold PBS containing 1% bovine serum albumin (BSA, fraction V). The organs were homogenized by grinding through a 100 μm nylon cell strainer (Biologix, Jinan, China).

### 2.5. Culture Media

For splenocyte cultivation, conditioned medium from a primary culture of mouse embryonic fibroblasts was obtained as described in [17]. Fibroblasts were cultured in 5% CO_2_ at 37 °C in RPMI-1640 medium (PanEco, Moscow, Russia) containing 10% inactivated fetal bovine serum (HyClone, Logan, UT, USA), 100 mg/L penicillin-streptomycin (PanEco, Moscow, Russia), 20 mM HEPES (PanEco, Moscow, Russia) and 10 mM 2-mercaptoethanol (Merck, Darmstadt, Germany). For inactivation, the serum was heated at 50 °C for 1.5 h. The medium was collected every 2–3 days. Before use, the conditioned medium was filtered (0.22 μm) and diluted 2-fold with unconditioned medium.

### 2.6. Study of the Proliferative Activity of Splenocytes Ex Vivo

The splenocyte suspension was seeded in a round bottom 96-well plate (JSC Medpolymer, St. Petersburg, Russia) at 600,000 cells per well. Lymphocyte proliferation was induced by concanavalin A (conA) (PanEco, Moscow, Russia) (5 μg/mL). Cells were cultured at 5% CO_2_ and 37 °C for 24 or 48 h. To assess lymphocyte proliferation, resazurin (Sigma Life Sciences, Oakville, ON, Canada) was added to the wells at a final concentration of 0.5 mM and incubated for another 5 h, after which the optical density in each well was measured using Tecan Infinite M200 (Tecan Group Ltd., Männedorf, Switzerland).

### 2.7. Cultivation of P815 Mastocytoma

P815 mastocytoma cells (K^d^D^d^, allogeneic tumor) were cultured in RPMI-1640 medium (PanEco, Moscow, Russia) supplemented with 10% fetal bovine serum and penicillin-streptomycin (PanEco, Moscow, Russia) at 37 °C, 5% CO_2_.

### 2.8. Evaluation of the Level of Antitumor Immune Response In Vivo

Male mice were immunized as described in [18]. Twelve days after immunization, spleen cells were stained with fluorescently labeled antibodies and analyzed using a flow cytometer CytoFLEX (Beckman Coulter, Brea, CA, USA). To study the secondary response, animals were re-injected with the tumor or PBS after 2 months, and splenocytes were isolated for analysis after 6 days.

### 2.9. Immunophenotyping

Spleen cells were incubated with Fc-block (CD16/CD32) and then covered with a mixture of the corresponding fluorescent conjugated antibodies to identify subpopulations of interest (Appendix A) following the manufacturer’s protocol. Analysis was performed on a flow cytometer CytoFLEX (Beckman Coulter, Brea, CA, USA). Dead cells were excluded based on scatter characteristics (Appendix A).

### 2.10. Isolation of T and B Lymphocytes

To separate T and B cells, 1 million splenocytes were centrifuged (300× *g*, 5 min), and the pellet was resuspended in 100 μL of buffer (2 mM EDTA, 0.5% BSA in PBS, pH 7.2). T and B cell populations were separated using MojoSort kits (BioLegend #480024 and #480051, San Diego, CA, USA).

### 2.11. Evaluation of Cell Distribution by Phases of the Spermatogenic Epithelium Cycle

Cell distribution was assessed as described in [19]. Briefly, the contents of testes were dissociated with collagenase IV, washed with DNAse solution, treated with 0.01% trypsin and passed through 70 µm nylon strainer. The suspended cells were fixed with 0.75% PFA, and the staining by propidium iodide, accompanied by permeabilization, was performed. The analysis was performed on a flow cytometer CytoFLEX (Beckman Coulter, Brea, CA, USA).

### 2.12. Sperm Motility Assessment

Sperm motility was assessed as described in [20]. The vas deferens was washed in Spermprep medium (PanEco, Moscow, Russia), and the resulting cell suspension was transferred to Universal IVF Medium (Origio, Malov, Denmark) coated with mineral oil. Video recording of sperm movement was performed using an Axiovert 200M inverted light microscope (Carl Zeiss AG, Oberkochen, Germany). Motility was calculated according to the WHO classification, assessing the ratio of progressively motile (PR), non-progressively motile (NP), and immotile sperm (IM) in the suspension.

### 2.13. Histology

Preparation of paraffin sections and histochemical staining of testes or ovaries with hematoxylin and eosin were performed as described in [19]. Fixed organs were preserved in Davidson’s solution for 7–8 h prior to embedding in Histomix (BioVitrum, St. Petersburg, Russia) and sections with a thickness of 4 μm were prepared using a semi-automatic rotary microtome (RMD-3000, MT Tochka, St. Petersburg, Russia). The area for sectioning was selected randomly. The obtained histological sections, stained with Mayer’s hematoxylin and eosin (BioVitrum, Russia), were mounted in Vitrogel mounting medium (BioVitrum, Russia). Images were obtained using a Nikon ECLIPSE Ti inverted microscope (Nikon Corporation, Tokyo, Japan).

### 2.14. Real-Time PCR

Total RNA from tissue samples was obtained using the ExtractRNA reagent (Eurogen, Moscow, Russia) following the manufacturer’s protocol. The isolated RNA was dissolved in nuclease-free water and stored at 4 °C. Reverse transcription was performed using the MMLV RT kit (Eurogen, Moscow, Russia). Prior to cDNA synthesis, RNA concentration was measured using a NanoPhotometer spectrophotometer (Implen, Munich, Germany) and its quality was assessed by electrophoresis in a 1.5% agarose gel. For the reaction, 0.5–2.5 μg of RNA was used. For real-time PCR, the ready-to-use qPCRmix-HS SYBR mixture (Eurogen, Moscow, Russia) was used. The amount of cDNA equivalent to 30 ng of treated RNA was used per reaction. Each sample was analyzed in three technical replicates. Hypoxanthine phosphoribosyl transferase (*Hprt1*) gene was used as a reference. The list of primers used is given in Appendix A.

### 2.15. Western Blot

Analysis of the protein composition of cells was performed using immunoblotting as described in [16]. Cells were lysed in RIPA buffer supplemented with protein inhibitor cocktail (Sigma-Aldrich, St. Louis, MO, USA). Total protein concentration was quantified by the Bradford method. Absorbance at 560 nm was measured with a CLARIOstar Plate Reader (BMG Labtech, Ortenberg, Germany). Proteins were separated by SDS-PAGE and transferred onto 0.22 μm nitrocellulose membrane (Bio-Rad, Hercules, CA, USA). After blocking with 5% skimmed milk, membranes were treated with primary antibodies and secondary antibodies following the manufacturer’s protocol. After incubation in the presence of secondary antibodies, chemiluminescence was detected using ECL (Bio-Rad Laboratories, CA, USA) on a transilluminator iBright FL1500 (Thermo Fisher Scientific, Waltham, MA, USA). The list of antibodies used is presented in Appendix A. The band intensity was calculated using ImageJ software, version 1.54n (National Institutes of Health, Bethesda, MD, USA).

### 2.16. Blood Collection from Mice

Blood was collected from the retroorbital venous sinuses of lightly anesthetized mice (2 mg/100 g body weight of Zoletil and Xyla each). Thirty microliters of 1 U/μL heparin were added per 100 μL of blood. Red blood cells were then lysed with red blood cell lysis buffer (0.8% NH_4_Cl, 0.1% NaHCO_3_, 0.0037% EDTA) and the remaining pellet was washed twice by centrifugation in PBS (PanEco, Moscow, Russia).

### 2.17. Statistical Data Analysis

Statistical analysis of the obtained data was performed using the GraphPad software package Prism 8.0.2 (GraphPad Software, San Diego, CA, USA). The normality of sample distribution was assessed using the Kolmogorov–Smirnov test. For non-normally distributed samples, the non-parametric Mann–Whitney U-test was used when comparing two groups. For multiple comparisons, one-way or two-way ANOVA was performed after pairwise comparisons (Sidak’s post hoc test). Differences with values of *p* < 0.05 (*), *p* < 0.01 (**), *p* < 0.001 (***) were considered statistically significant.

## 3. Results

### 3.1. Expression of the Mutant Variant PolG^D257A^ Does Not Lead to Significant Changes in the Muscular and Reproductive Systems of Mice

We have previously shown that in mice CMV-Cre/PolG^mut^ at the age of 3 months, a deterioration in physical performance is observed [16]. Measurement of the dynamics of endurance and muscle strength in mice aged 4 to 10 months did not reveal significant differences between CMV-Cre/PolG^mut^ and control animals (Figure 1A). In addition, we did not find significant differences in the expression of mitophagy genes (Appendix A) in the muscles of CMV-Cre/PolG^mut^ mice. However, in CMV-Cre/PolG^mut^ mice at the age of 10 months, body weight was reduced by 1.5 times compared to control CMV-Cre (Figure 1B).

It is known that mitochondrial dysfunctions can lead to fertility disorders in both females and males [11,21,22]. Analyses of the composition of the spermatogenic epithelium and sperm motility of male CMV-Cre/PolG^mut^ mice at the age of 10 months did not reveal significant pathologies (Figure 1C,D). In CMV-Cre/PolG^mut^ males a slight (*p* > 0.05) decrease in the content of tetraploid spermatocytes (at the stage of meiosis I) and an increase in the proportion of elongated spermatids (cells at stage 1n) relative to control animals CMV-Cre (Figure 1D) were observed. Unfortunately, these differences were statistically insignificant. Moreover, in male CMV-Cre/PolG^mut^ mice the proportion of immotile spermatozoa was increased by an average of 30% (Figure 1E). These data indicate the influence of mutant *mPolg* expression on mature spermatozoa, but not on the maturation of germ cells in male mice.

A study of the functional state of the female reproductive system in females aged 7.5 months showed that expression of the PolG^D257A^ variant does not lead to disruption of oogenesis or a decrease in ovarian reserve (Figure 1F).

### 3.2. Expression of a Mutant Variant of the mPolg Gene Leads to Significant Disturbances in the Immune System

It is known that the functional state of mitochondria affects the processes of maintenance, activation and differentiation of T cells, especially CD8+ T lymphocytes [23,24,25], therefore, genetically determined mitochondrial dysfunction can lead to significant changes in the T-cellular component of the immune system. We found that the expression of the mutant variant *mPolg* leads to accelerated thymus involution: in double transgenic CMV-Cre/PolG^mut^ mice at the age of 4 months, the thymus is reduced to the point of disappearance compared to mice of the activator CMV-Cre line (Figure 2A). Limper et al. demonstrated that mice carrying constitutive PolG^D257A^ mutations, both heterozygous and homozygous, exhibit a decreased proliferative potential of thymocytes at the double-negative (DN) stage and a developmental block at the DN3–DN4 transition [12]. The authors attributed this effect to the requirement for active mitochondrial respiration during the highly proliferative DN3–DN4 phase of thymocyte development. At the same time, in CMV-Cre/PolG^mut^ mice, an accumulation of immature thymocytes occurs in the DN population (Figure 2B), which is possibly associated with an arrest at the DN4 stage (Figure 2C).

Despite the early involution of the thymus and the tendency to decrease in overall cellularity of thymocytes (Figure 2D), no decrease in cell numbers was observed in the peripheral lymphoid organs of double transgenic CMV-Cre/PolG^mut^ mice compared to control animals. We found that at the age of 6 months the relative number of cytotoxic CD8+ T lymphocytes increases significantly in CMV-Cre/PolG^mut^ transgenic mice—for comparison, the CD4+/CD8+ ratio among CD3+ cells in CMV-Cre mice is 1.61, while in CMV-Cre/PolG^mut^ it is 0.96 (Figure 2E). Analysis of CD8+ T-cell subpopulations in CMV-Cre/PolG^mut^ mice revealed a significant decrease in the proportion of naïve T lymphocytes together with an increase in the proportion of cells with the effector phenotype compared to CMV-Cre mice (Figure 2F, Table 1). However, we did not find such changes in young CMV-Cre/PolG^mut^ mice (2–5 months) (Figure 2E).

Primary allogeneic response to P815 mastocytoma cells in vivo in double transgenic mice CMV-Cre/PolG^mut^ does not differ from control mice, while the secondary response in CMV-Cre/PolG^mut^ mice decreased by 2.5-fold compared to control mice (Figure 3A). In addition, splenocytes of CMV-Cre/PolG^mut^ mice demonstrated reduced (almost 2-fold) proliferative activity compared to splenocytes from CMV-Cre mice only 48 h after ConA treatment, but not after 24 h in vitro (Figure 3B). Taken together, the data from the experiments both in vivo and in vitro indicate a decrease in the activity of T-lymphocytes with mtDNA mutations, which is characteristic for lymphocyte exhaustion [26].

The processes of lymphocyte activation and differentiation largely depend on the function of signal transducer and activator of transcription (STAT) family transcription factors. We found that in the T and B cells of the spleen of CMV-Cre/PolG^mut^ mice there is a decrease in both the native (inactive) form of Stat5 and its active form, phosphorylated at tyrosine 694—p-Stat5 Y694 (Figure 3C,D). The most well-known function of p-Stat5 Y694 is the regulation of the immune response. Indeed, in both T and B lymphocytes of CMV-Cre/PolG^mut^ mice we detected lower levels of the proinflammatory factor NFκB p65 (Figure 3D), compared with WT. Wild-type mice were used as a control, as CMV-Cre has been shown not to affect the T-cell immune response and Stat5 dependent pathways [27]. Interestingly, B lymphocytes and, to a lesser extent, T cells showed elevated levels of the anti- and proapoptotic proteins Bcl-w and Bad, respectively (Figure 3D), which is common in mitochondrial dysfunction. The unaltered glycolysis in the presence of mitochondrial genomic instability may be explained by accelerated T lymphocyte exhaustion: we found that CMV-Cre/PolG^mut^ mice had no changes in the levels of the glycolytic enzyme GAPDH in either lymphocyte type compared to control mice.

### 3.3. Blood Cells Expressing mPolg^mut^ Are Gradually Eliminated from Bloodstream

When using the CMV-Cre activator line, the induction of transgene expression occurs at the early stages of preimplantation development, thus the mutant variant of *mPolg* is expressed in all cells of CMV-Cre/PolG^mut^ animals. On the other hand, in ROSA-Cre/ER^T2^/PolG^mut^-GFP, the excision of the STOP cassette occurs upon tamoxifen treatment. Due to the limitations of this induction system, excision may not occur in all cells of the body [15]. To assess the ratio of cells expressing and not expressing the transgene in the same organism, we activated transgene expression in ROSA-Cre/ER^T2^/PolG^mut^-GFP mice by injecting tamoxifen into adult mice aged 6–9 months and studied the change in the relative number of GFP-positive cells in the peripheral blood within a time course of six weeks after transgene activation. It appears that in two weeks after induction, GFP+ cells constituted just 20–25% of the peripheral blood cell counts in ROSA-Cre/ER^T2^/PolG^mut^-GFP mice and the relative number of GFP+ cells decreased with time (Figure 3E). The observed effect may indicate the substitution of cells expressing the transgene by cells in which the STOP cassette has not been excised. This hypothesis can be supported by the decreased proliferative activity of *mPolg^mut^* expressing thymocytes (Figure 3B).

## 4. Discussion

Mitochondrial diseases associated with mtDNA mutations are a common group of inherited metabolic disorders, the treatment of which is complicated by the significant variability of such mutations [28]. Mice with a mutant gene *mPolg* are a well-studied model of mitochondrial diseases and the progeroid phenotype [3,4,5]. The most popular model is the one with the constitutive mutation PolG^D257A^; however, the main drawback in using these models in the progressive accumulation of mtDNA mutations, which severely restricts animal breeding due to reduced fertility and viability. Moreover, these mutations are highly heterogeneous, differing substantially between individual animals; as a result, experimental cohorts often have significant inter-individual variability, making it challenging to acquire robust statistical power. To implement a more relevant model, we created two strains of transgenic mice with inducible expression of the mutant variant PolG^D257A^. In one line, for ease of detection of the transgene, the *Gfp* gene (PolG^mut^-STOP-GFP) [16] was inserted after *PolG^mut^* separated by internal ribosome entry site (IRES), while the transgenic line PolG^mut^-STOP-pKB2 does not contain a reporter gene as part of the transgene (Appendix A). The main advantage of the model used is that site-specific recombination at *LoxP* sites and excision of the STOP cassette occurs early in embryonic development or, in the case of using the ROSA-Cre/ER^T2^ activator line, after induction [15,29]. As a result, expression of the mutant *mPolg* gene in each individual organism begins de novo. Our model may be relevant for the study of clinical cases, since heteroplasmy is usually accompanied by a gradual accumulation of mutations and the development of pathological symptoms [30].

In young CMV-Cre/PolG^mut^ at the age of 3 months, a deterioration in physical parameters was found compared to control mice [16], however, in a dynamic study of CMV-Cre/PolG^mut^ from the age of 4 to 10 months, no differences were found (Figure 1A). This effect can be explained by the fact that physical parameters were analyzed throughout the life of the animals, which led to an improvement in their physical condition [9,31]. At the same time, CMV-Cre/PolG^mut^ had a 1.5-fold decrease in body weight compared to control CMV-Cre (Figure 1B), which partially coincides with the data obtained in a study of the effects of tissue-specific mutation PolG^D257A^ in muscle cells [8].

It has been previously shown that the expression level of mitophagy markers in the culture of CMV-Cre/PolG^mut^ mouse embryonic fibroblasts was significantly increased [16]. We did not find significant differences in the expression of these genes in the muscles of adult CMV-Cre/PolG^mut^ mice, although we observed a tendency towards a lower level of expression of their genes (Appendix A). We suggest that this phenomenon is explained by the presence of compensatory mechanisms in the body, resulting in the elimination of either the damaged mitochondria themselves or the cells containing them. In a study on ROSA-Cre/ER^T2^/PolG^mut^-GFP mice we showed that the percentage of GFP+ cells in the blood of mice decreased over time, which confirms our hypothesis (Figure 3E).

The D257A mutation introduced into the mouse genome is associated with premature loss of fertility and placental insufficiency in young females [21], as well as atrophy of the spermatogenic epithelium, vacuolization of the seminiferous tubules and an increase in the number of Leydig cells in adult males [11,22]. In our studies, the spermatogenic epithelium of 10-month-old CMV-Cre/PolG^mut^ male mice did not have any pronounced pathologies (Figure 1C–E). A slight decrease in the number of tetraploid spermatocytes in favor of elongated spermatids (Figure 1D) is probably associated with the accumulation of the latter in the apical part of the tubule due to insufficient mitochondrial activity [32], however, these differences were not statistically significant. Strikingly, in CMV-Cre/PolG^mut^ mice the number of immotile spermatozoa was significantly increased (Figure 1E). Mutations of mtDNA in PolG^D257A^ mice can vary significantly from generation to generation, and their high levels in young animals may be associated with metabolic disturbances in breeding males. In our study, we demonstrated that even with the simultaneous expression of normal *mPolg* and the mutant form, an accumulation of low-motility sperm occurs. This effect can be explained by the need for active mitochondria for normal activity of the sperm flagellum [33]. These data are consistent with our understanding of the reproductive difficulties of mice with a genomic PolG^D257A^ mutation. A study of the functional state of the female reproductive system in females aged 7.5 months showed that the expression of PolG^mut^ de novo does not lead to disruption of oogenesis (Figure 1F). These results can be explained by the fact that oocytes are formed early in embryonic development, when the number of accumulated mutations in mtDNA is not yet sufficient to influence the development of female germ cells. We suggest that the absence of pathologies in the organs of the reproductive and muscular systems of mice may be associated with a lower rate of mutation accumulation, since in addition to the mutant variant of *mPolg*, the normal variant is also expressed.

The most striking changes in CMV-Cre/PolG^mut^ mice were those in the immune system, which is typical for PolG^D257A^ mutator mice [13,34,35]. Accelerated thymus involution in the mice studied was detected as early as 4 months of age (Figure 2A). The reasons for this phenomenon remain controversial: accelerated thymus atrophy may be associated with impaired migration of early precursors from the bone marrow, or caused by defects in the process of intrathymic T-cell selection. Thymic involution is a normal age-related phenomenon in mice continuing through adulthood. However, accelerated thymus involution and changes in the CD8+ T cell compartment are hallmarks of immune aging (also known as “inflammaging”), or immunosenescence [36]. The decline in naive T cell production is closely linked to an increase in the proportion of effector T cells, particularly cytotoxic CD8+ T cells, whose persistent activation can contribute to chronic inflammation and tissue dysfunction [37,38,39]. The double-negative (DN) population of T lymphocytes represent the earliest stage in T cell development within the thymus, characterized by the absence of both CD4 and CD8 surface markers. Among the DN stages, DN4 are critical transitional phases: DN4 thymocytes represent the stage immediately following β-selection, where cells proliferate and begin to express both CD4 and CD8, transitioning to the double-positive (DP) stage [12]. The transition to the DP stage appears to require considerable cellular energy, necessitating robust mitochondrial metabolism to sustain intensive proliferation and differentiation. Analysis of the population composition of cells in the thymuses revealed that already at the age of 4 months, CMV-Cre/PolG^mut^ mice demonstrated an accumulation of cells in the DN4 occurs (Figure 2B,C), which is consistent with the data obtained previously [12]. Although changes in the thymus occur already in young mice, it is only at the age of 6 months that a shift in the balance of T-lymphocytes towards cytotoxic (CD8+) effector T-lymphocytes and a decrease in the relative number of naive T-cells in the secondary lymphoid organs occur (Figure 2D–F). At the same time, modeling of the immune response in vivo and ex vivo revealed a decrease in the proliferative activity of CMV-Cre/PolG^mut^ lymphocytes by almost 2 times compared to CMV-Cre mice (Figure 3A,B), which may be caused by T-cell exhaustion, but this conclusion requires further research. The observed T-cell phenotype of PolG^mut^ mice validated by flow cytometry coincides with the concept of age-related changes in the immune system of mice and humans [38,39,40].

The investigation of the molecular causes underlying the reduced immune response in CMV-Cre/PolG^mut^ mice, conducted in the context of an increased proportion of effector cytotoxic T lymphocytes, revealed a decrease in the production of the critical transcription factor Stat5 in both T and B lymphocytes (Figure 3C,D). It is known that Stat5 is an essential transcription factor for all cells of the immune system (more details in [41]). Stat5 phosphorylation occurs in response to the binding of cytokines such as IL-2, IL-7 or IL-15 to the corresponding receptors on the surface of T lymphocytes, which is necessary for maintaining their effector functions [42,43]. It has been shown that in Stat5 knockout mice, T lymphocytes do not differentiate [44], which is similar with our findings. Additionally, we found that in CMV-Cre/PolG^mut^ mice levels of the NF-kB p65 form, a transcription factor necessary for the transcription of proinflammatory cytokines, are slightly reduced [45]. These effects may be attributable to metabolic exhaustion. Cells lacking active mitochondria have diminished energy reserves, which can suppress essential immune signaling pathways such as Stat5 and NF-kB activation [46,47]. Thus, compromised mitochondrial health in CMV-Cre/PolG^mut^ mice likely contributes to impairment of cytokine signaling and diminished immune responses observed in these animals.

Thus, expression of the mutant variant PolG^D257A^ de novo leads to accelerated involution of the thymus, an increase in the proportion of T-lymphocytes with an effector phenotype in the splenocyte population, a decrease in the level of the secondary immune response and the proliferative potential of lymphocytes in vitro. However, the molecular mechanisms of this phenomenon remain unknown, and their identification requires further studies on models with tissue-specific expression of the mutant *mPolg.* While it is well established that the PolG mutation leads to dysfunction across multiple organ systems, our study suggests that mtDNA mutations primarily impair the immune system in mice, with a predominant effect on the T-cell compartment, at the stage of active proliferation of differentiating T cells (DN4). The obtained data are consistent with the concept of the key role of mitochondrial activity in the differentiation, proliferation and functional activity of immune system cells and, consequently, accelerated aging of the immune system. Recent studies have shown that patients with mtDNA mutations exhibit immunological dysfunction, including accelerated immune aging and increased inflammation [48,49]. Thus, our PolG^mut^ model recapitulates some of the features of immune aging due to mitochondrial diseases reported in individuals with mtDNA mutations, confirming its translational significance while acknowledging the need for further clinical validation.

## 5. Conclusions

A comprehensive study of the phenotype of de novo expression PolG^mut^ mice revealed moderate manifestations of mitochondrial dysfunction, even in mice at 10 months of age, primarily affecting body weight and immunity, but not leading to fatal morphological consequences. This may be due to the retention of the wild-type allele of the *mPolg* gene, which slows the accumulation of a critical level of mtDNA mutations. Despite the fact that the immunity of mice and humans differs significantly, mitochondrial biogenesis and function are sufficiently conserved processes, so this model can be used to develop treatment methods or therapies for mitochondrial diseases. We believe that significant differences between the studied model of mitochondrial dysfunction and previously published models are due to the induction of expression of the mutant variant of PolG^D257A^ de novo in an individual organism and the presence of wild-type *mPolg* gene expression. However, we suggest that the less aggressive phenotype makes our PolG^mut^ model more relevant for research into treatments for mitochondrial diseases and premature aging associated with immune disorders only.

## Figures and Tables

**Figure 1 life-15-01863-f001:**
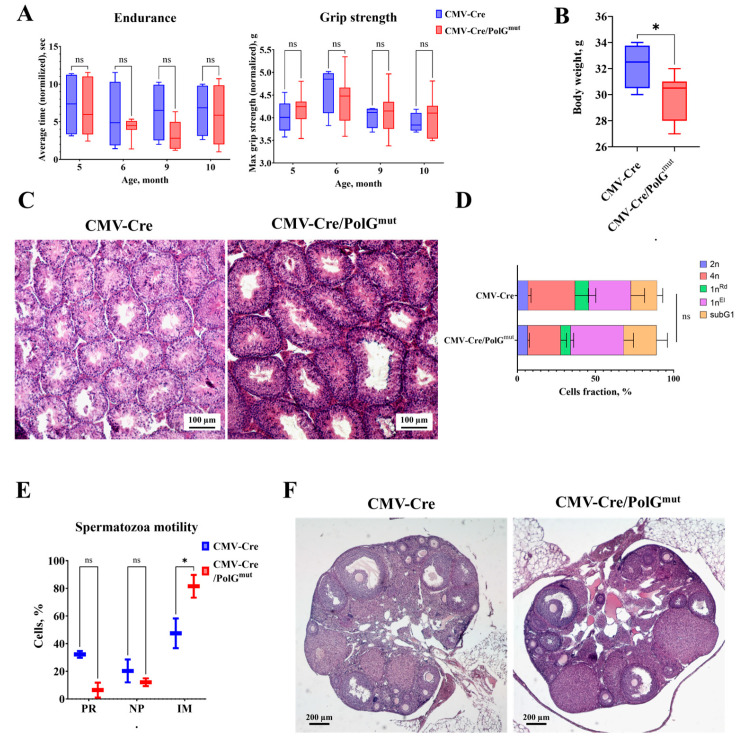
Expression of the mutant variant of *mPolg* does not lead to significant changes in the muscular and reproductive systems of mice. (**A**) Dynamics of changes in endurance and grip strength in the experimental (CMV-Cre/PolG^mut^-GFP) (*n* = 10) and control (CMV-Cre) (*n* = 4) groups (Two-way ANOVA, Sidak post-hock test). (**B**) Comparison of body weight of mice in the experimental (CMV-Cre/PolG^mut^-GFP) (*n* = 10) and control (CMV-Cre) (*n* = 4) groups (Mann–Whitney test). (**C**) Cross-section of testis of control (CMV-Cre) and transgenic (CMV-Cre/PolG^mut^-GFP) mice, 100× magnification, hematoxylin and eosin. (**D**) Ratio of spermatogenic epithelial cell subpopulations in transgenic (CMV-Cre/PolG^mut^-GFP) and control (CMV-Cre) mice at 10 months of age (Mann–Whitney U-test). (**E**) Ratio of the number of motile (PR), non-progressively motile (NP) and immotile (IM) spermatozoa in transgenic (CMV-Cre/PolG^mut^-GFP) and control (CMV-Cre) mice (Mann–Whitney U-test). (**F**) Cross-section of the ovary of a control (CMV-Cre) and transgenic (CMV-Cre/PolG^mut^-GFP) mouse at the age of 7.5 months, magnification 400×, hematoxylin and eosin. *: *p* < 0.05. ns: not significant.

**Figure 2 life-15-01863-f002:**
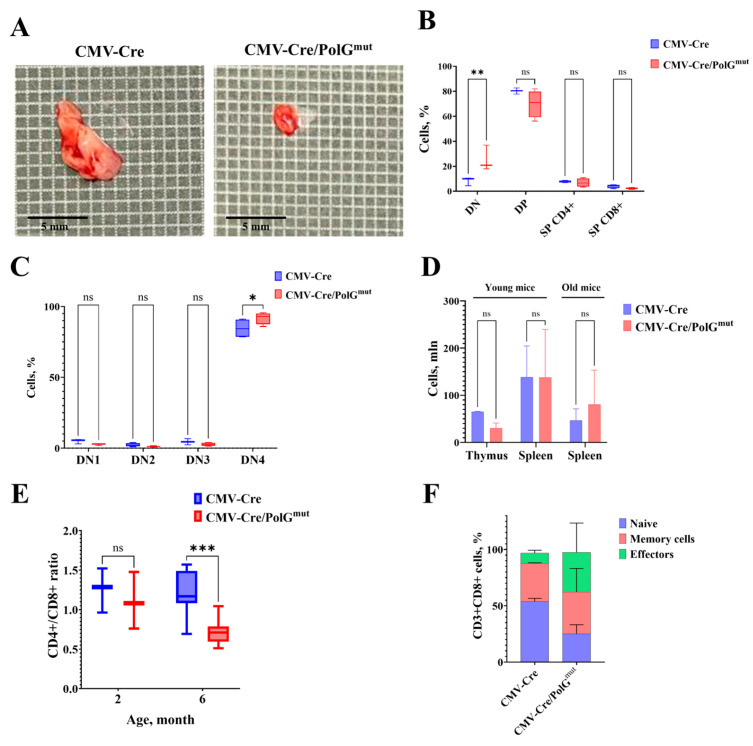
Expression of the mutant variant of *mPolg* results in a decrease in the proportion of naive T lymphocytes in favor of effector T cells. (**A**) Thymuses of control (CMV-Cre) and transgenic (CMV-Cre/PolG^mut^-pKB2) mice (4 months). (**B**,**C**) Analysis of population composition in the thymuses of mice at the age of 4 months. *n* = 3, mean ± SD. DN—double negative (CD4−CD8−), DP—double positive (CD4+CD8+), SP—single positive (CD4+CD8− or CD4−CD8+), DN1 (CD44+CD25−), DN2 (CD44+CD25+), DN3 (CD44−CD25+), DN4 (CD44−CD25−). (**D**) Total cellularity in the thymus and spleen of young (2–5 months) and aged (6–8 months) CMV-Cre and CMV-Cre/PolG^mut^-pKB2 mice. *n* = 3, mean + SD. (**E**) CD4+/CD8+ T cell ratio in spleens of CMV-Cre/PolG^mut^-pKB2 mice (Mann–Whitney U test, *n* = 4). (**F**) Relative number of CD8+ naive T cells of the (CD62L+CD44−) phenotype is decreased in CMV-Cre/PolG^mut^-pKB2 mice at 6 months compared to control animals (*n* = 4, two-way ANOVA). *: *p* < 0.05. **: *p* < 0.01. ***: *p* < 0.001. ns: not significant.

**Figure 3 life-15-01863-f003:**
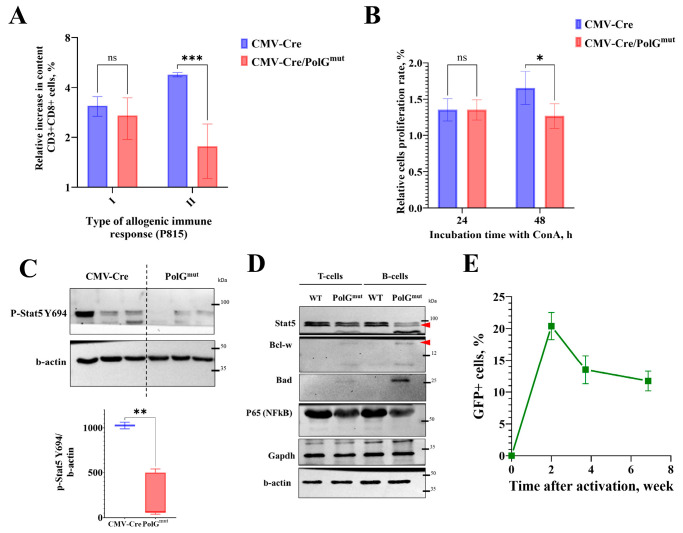
In CMV-Cre/PolG^mut^-pKB2 mice the secondary antitumor immune response is reduced. (**A**) Relative number of effector cytotoxic T lymphocytes (CD8+CD62L−CD44+) in spleens of control (CMV-Cre) and transgenic (CMV-Cre/PolG^mut^-pKB2) mice in response to primary and secondary stimulation with allogeneic P815 tumor. Cell count increment was calculated as ratio of percentage of CD8+ T cells in tumor-challenged mice to percentage of CD8+ T cells in PBS-injected mice. Primary response: *n* = 4, secondary response: *n* = 3. (**B**) Change in optical density of resazurin in spleen cell culture stimulated with ConA relative to intact cell culture after 24 h (*n* = 6) or 48 h (*n* = 5) (two-way ANOVA). The optical density of the medium without cells was used as the background value, and the optical density of cells without ConA stimulation was taken as 100%. (**C**) Western blot analysis of spleens from transgenic CMV-Cre/PolG^mut^-pKB2 and control CMV-Cre mice (**top**) and analysis of p-Stat5 Y694 band intensity relative to the control protein β-actin (**bottom**) performed using ImageJ software, version 1.54n (*t*-test, *n* = 3). (**D**) Western blot analysis of splenocytes from CMV-Cre/PolG^mut^-pKB2 and control animals (WT); the red arrow marks specific bands. (**E**) The proportion of GFP+ cells in the peripheral blood of ROSA-Cre/ER^T2^/PolG^mut^-GFP mice 2 weeks after the start of mouse activation (4 months) (mean ± SEM). *: *p* < 0.05. **: *p* < 0.01. ***: *p* < 0.001. ns: not significant.

**Table 1 life-15-01863-t001:** Proportion of cells in CD8+ T cell subsets in the spleens of 6-month-old mice (mean ± SD).

Types of CD8+ T-Cells	CMV-Cre, %	CMV-Cre/PolG^mut^-pKB2
Naïve cells (CD62L+CD44−)	50.50 ± 7.12	25.79 ± 6.98
Memory cells (CD62L+CD44+)	32.99 ± 1.81	37.40 ± 17.99
Effectors (CD62L−CD44+)	13.26 ± 8.52	33.68 ± 22.75

## Data Availability

The original contributions presented in this study are included in the article/Appendix A. Further inquiries can be directed to the corresponding author.

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
