# Peer review of "Ubiquitous Expression of *mPolg^mut^* Leads to the Accumulation of Cytotoxic CD8+ T Lymphocytes in Young Mice"

_life, 2025, doi:10.3390/life15121863_

Round 1

Reviewer 1 Report

Comments and Suggestions for Authors

This work focuses on investigation into the relationship between mitochondrial DNA (mtDNA) mutations and age-related immune system dysfunction using inducible transgenic mouse models expressing mutant DNA polymerase γ (PolG^mut). The study is conceptually strong and offers a potentially valuable model for dissecting mitochondrial contributions to aging. My comments to improve the manuscript before publication are as follows:

  1. Abstract: The distinction between the new inducible PolG^mut lines and the conventional germline PolG^mut mice should be emphasized earlier to immediately convey the study’s innovation.
  2. The rationale for developing inducible PolG^mut models is compelling. However, the abstract should more clearly articulate why de novo activation at different developmental stages is critical for understanding mitochondrial aging mechanisms.
  3. Highlight how this approach advances beyond the existing PolG^mut model, particularly regarding disentangling primary mtDNA mutation effects from secondary compensatory responses.
  4. The identification of thymus involution and CD8+ T cell alterations as key outcomes is interesting and warrants discussion of how this immune phenotype relates to systemic aging processes.
  5. While the link between mtDNA mutation accumulation and T-cell dysfunction is proposed, however, elaborate on whether observed immune phenotypes arise from energy deficits, ROS generation, or altered signaling pathways.
  6. The selective vulnerability of thymic and peripheral immune cells compared to other tissues is intriguing. Possible explanations (e.g., high proliferative rate, dependence on mitochondrial metabolism) should be outlined.
  7. Discuss whether accelerated thymic involution mimics physiological aging or represents an exaggerated pathological model.
  8. More information is needed about the inducible system used (e.g., Cre-loxP, Tet-On/Tet-Off) and how induction timing was controlled.
  9. Indicate whether the observed T-cell phenotypes were validated using histology, flow cytometry, or functional assays.
  10. Include mention of controls, particularly whether wild-type or non-induced transgenic animals were used for baseline comparison.
  11. In interpretation, additional context comparing these findings to age-related mtDNA mutation patterns in human tissues would strengthen translational relevance.
  12. The conclusion that mtDNA mutations “primarily lead to the inability to maintain normal T-cell function” should be framed as a working hypothesis rather than a definitive statement.
  13. Consider whether the absence of pathology in other organs reflects lower induction efficiency, delayed onset, or inherent tissue resilience.
  14. The following studies are suggested to evaluate and add to the literature review of the manuscript: https://doi.org/10.1002/advs.202308711, https://doi.org/10.3389/fimmu.2024.1383255, https://doi.org/10.1080/21645515.2023.2263229
  15. Figure 1 data is too dense. Please make sure that all data is readable and appropriate flow. 

Author Response

Dear reviewer,

The authors express their gratitude for the positive evaluation of the article and for the important comments they made to improve it.

Reviewer 2 Report

Comments and Suggestions for Authors

Major Comments

- When describing their results, the authors should follow the figure sequence consistently. Avoid making the reader move back and forth to locate the figures. In addition, Figure 2F is not mentioned anywhere in the text.  Same issue is again in the discussion part between Fig2 and Fig 1.

-The figure2 legend is not matched with figure labeling of J and K.  In addition, there is wrong figure inserted in the text on the line 307 on page 8.  It happened in discussion part as well.

-Authors should check the entire paper to see if all inserted figures match the corresponding sentences.

Minor Comments

-Correct “C57BI/6” to “C57BL/6” on line 64, page 2.

 -Correct “+37 °C” to “37 °C” on line 92, page 3. A similar temperature notation error occurs on line 151, page 4 — correct it as well.

 -Provide the full gene name for Hprt.

 -Change “Figure 2D” to “Figure 2F” on line 243, page 6.

 -Spell out “STAT” on line 281, page 8.

 -Clarify the following sentence on lines 292–293, page 8: “In adult ROSA-Cre/ERT2/PolGmut-GFP mice, after induction of transgene expression, lymphocytes expressing the transgene are gradually replaced by lymphocytes without one (???)”

 -Remove or translate the Russian words found on line 314, page 8.

 -Spell out “IRES” on line 323, page 9.

 -Correct “C57BI/6” to “C57BL/6” in Table A1.

Comments on the Quality of English Language

Not applicable

Reviewer 3 Report

Comments and Suggestions for Authors

This manuscript describes two new mouse models with inducible expression of a mutant mitochondrial DNA polymerase γ (PolG^D257A), designed to study de novo mtDNA mutation accumulation. The authors demonstrate that the most pronounced phenotype is accelerated thymus involution and accumulation of cytotoxic CD8⁺ effector T cells, whereas other organ systems (muscle and reproductive) remain largely unaffected. They conclude that immune dysfunction is a primary effect of mitochondrial genomic instability, contributing to premature immune aging.

The topic is scientifically relevant and fits the journal’s scope in mitochondrial biology and aging research. The study presents a technically sound genetic approach and interesting immunological observations. However, several aspects of experimental design, data analysis, and mechanistic interpretation require major revisions before publication.

Major concerns:

The authors use CMV-Cre and ROSA-Cre/ERT2 lines for transgene activation, but Cre expression and tamoxifen treatment themselves can cause off-target or inflammatory effects. Include Cre-only and tamoxifen-only control groups for all phenotypic assays or alternatively provide literature or prior data demonstrating no significant immune impact from these factors.

While the inducible model is novel, the study does not clearly compare its mtDNA mutation dynamics to classical PolG^D257A “mutator” mice. Discuss or quantify differences in mutation accumulation, lifespan, or phenotype severity relative to published models (e.g., Trifunovic et al., Nature, 2004; Kujoth et al., Science, 2005).

The conclusion states that the model is “clinically relevant” for studying human mitochondrial disease and premature aging, yet no human data or validation are presented. Temper this claim, or add discussion referencing comparable immune dysfunctions in human PolG patients.

Minor Concerns

Replace untranslated Russian text (“митохондриальных заболеваний”) with English equivalents.

Formatting: Correct inconsistent capitalization (e.g., “PolGD257A variant” vs. “PolGmut”) and italicize gene names.

Language: Several sentences in the Abstract and Discussion are repetitive or grammatically awkward; careful editing for conciseness is advised.

Ethics statement: The ethical approval section is present but should specify the approval number and date in the Methods, not just at the end.

Sex differences: Clarify whether both sexes were included in immune assays; this is relevant for T-cell studies.

Comments on the Quality of English Language

Manuscipt still contain russian text

Author Response

(The authors gave the same response as above.)

Round 2

Reviewer 1 Report

Comments and Suggestions for Authors

All comments addressed/resolved. No more comments 

Reviewer 2 Report

Comments and Suggestions for Authors

All issues listed have been resolved.

Reviewer 3 Report

Comments and Suggestions for Authors

All concerns are answered. Manuscript can be published